



# Surficial sediment remobilization by shear between sediment and water above tsunamigenic megathrust ruptures: experimental study

Chloé Seibert[1], Cecilia McHugh[2,1], Chris Paola[3], Leonardo Seeber[1], James Tucker[3]

[1]Lamont-Doherty Earth Observatory of Columbia University, Palisades New York, USA
5 [2]Queens College, City University of New York, School of earth and Environmental Sciences, New York 11367, USA
[3]Department of Earth and Environmental Sciences, St Anthony Falls Laboratory, University of Minnesota, Minneapolis, MN, USA

*Correspondence to*: Chloé Seibert (cseibert@ldeo.columbia.edu)

**Abstract.** Large megathrust earthquakes that rupture the shallow part of the interface can cause unusually large co-seismic 10 displacements and tsunamis. The long duration of the seismic source and high upper-plate compliance contribute to large and protracted long-period motions. The resulting shear stress at the sediment/water interface in, for example, the Mw9.0 2011 Tohoku-Oki earthquake, could account for the surficial sediment remobilization identified on the outer margin. Through physical tank experiments, we test this hypothesis by exploring shear between sediment and water, interactions between high and low frequency seismic waves, and sediment properties (chemistry, grain size, water content and salinity). Our results show 15 that low-frequency motion during a 2011-like earthquake can entrain several centimeters of surficial sediment and that entrainment can be enhanced by high-frequency vertical oscillations. These experiments validate a new mechanism of co-seismic sediment entrainment in deep-water environments.

## 1 Introduction

All earthquakes in the ≥M9.0 class originate from subduction megathrusts boundaries. Despite their high tsunamigenic risk, 20 such earthquakes are still poorly understood, as demonstrated by the last two earthquakes in this class, the Mw9.3 2004 Sumatra-Andaman and the Mw9.0 2011 Tohoku-Oki events (Lay, 2015). They ruptured the shallowest portion of the megathrust, which had been considered aseismic and was responsible for their catastrophic tsunamis. Five events of this size have been experienced in the last century, but the lack of data about them leads to two critical needs for constraining global hazard: 1) identifying subduction boundaries capable of producing very large earthquakes; and 2) determining the recurrence 25 of these events.

Characterization of earthquake event deposits in the offshore sedimentary record has extended earthquake catalogs into pre-history and improved seismic hazard estimations (e.g. Goldfinger et al., 2003, 2012; Pouderoux et al., 2014; Ratzov, et al., 2015; Usami et al., 2018; Seibert et al., 2024). Current work aims to relate distinctions among event deposits with earthquake



characteristics such as magnitude and source locations (Goldfinger et al., 2013; Moernaut et al., 2017; Van Daele et al., 2019;
McHugh et al., 2020; Howarth et al., 2021).

Understanding event deposits starts with sediment entrainment. The premise in sediment dynamics has been that the bed is
fixed, and the fluid moves relative to it, creating shear stress that entrains sediment. Seafloor motion from large subduction
earthquakes can be so large that alone could move the bed relative to ambient water fast enough to entrain sediment (Gomberg,
2018). The implication for paleoseismology is that large megathrust earthquakes could mobilize sediment in places where that
would otherwise be unlikely, creating distinctive sedimentary signatures.

As one of the most instrumented in history, the Tohoku-Oki earthquake and tsunami provided detailed ground-truths to relate
with specific sedimentary signatures, including: a persistent deep sediment suspension (Noguchi et al., 2012; Oguri et al.,
2013); a tsunami-remobilized sediment possibly sourcing turbidity currents to the upper slope (Arai et al., 2013; Toyofuku et
al., 2014; Tamura et al., 2015; Usami et al., 2017); large slumps at the trench (Kodaira et al., 2012; Strasser et al., 2013);
turbidity flows originating on the slope and trench (Ikehara et al., 2014, 2016; Molenaar et al., 2019); and surficial sediment
remobilization over ~100's of km$^2$ including the mid-slope terrace and trench (McHugh et al., 2016, 2020). McHugh et al.
(2016) documented this mechanism using short-lived radioisotopes. They highlighted 2011 event deposits consisting of a basal
turbidite overlain by a homogeneous muddy flow deposit (from 3 to 200 cm-thick). These mud layers are enriched in excess
$(xs)^{210}Pb$ that has not decayed and is in steady state, requiring a source from the upper few cm of sediment. This source must
also be widespread, to account for the thickness of the deposits. In addition to the Japan Subduction zone, surficial sediment
remobilization has been documented along the Nankai accretionary prism (Ashi et al., 2014) and in Chilean lakes (Moernaut
et al., 2017; Molenaar et al., 2021).

In this paper, we focus on surficial entrainment described in Japan (McHugh et al., 2016, 2020) and on large-amplitude long-
period seismic motion on the outer upper plate as a possible cause. The spectrum of seismic waves broadens toward long
periods for larger earthquakes, particularly for ones that include the slow-rupturing shallowest part of the megathrust (e.g. Lay,
2015). This suggests that retrieving a fingerprint of long-period seismic motion in sediment records may open opportunities to
differentiate the sedimentary signatures of the largest earthquakes.

From published results about the 2011 M9.0 rupture, we estimate the co-seismic long-period motions on the Japan slope-trench
margin, and we report on laboratory experiments testing their potential for entraining sediments. We believe these are the first
experiments on entrainment that combine high-frequency shaking of the sediment with shear between sediment and water that
captures the effects of long period motions. We also provide initial observations on how sediment properties (grain size,
mineralogy, water content and salinity) affect surficial sediment remobilization.

## 2 Conceptual model

The Tohoku-Oki earthquake ruptured the entire brittle depth range of the Japan erosional subduction boundary. From its
nucleation, this rupture propagated both down dip into the mantle and up-dip to the trench (Fig. 1A). The co-seismic





displacement increased up-dip, reaching ~60 m at the trench (Lay et al., 2011a; Fujiwara et al., 2011). Based on observations in Japan and elsewhere, Lay et al. (2012) subdivided the subduction interface in depth domains that radiate seismically with distinct spectral characteristics (Fig. 1A). In the shallowest of these megathrust domains, the rupture radiates preferentially at low frequencies (Lay et al., 2011b). This rupture domain underlies the outer part of the margin, where low rigidity (Fig. 1) and

low seismic velocities (Kodaira et al., 2017) independently characterize the upper plate. Seismic attenuation is proportional to frequency and in the outer upper plate it is likely to be high due to the presence of fluids rising from the subduction channel (e.g. Escobar et al., 2019), contributing further to a spectral shift toward the long period at the sea floor. An important contribution to long-period oscillations of the seafloor, however, may come from the resonance coupling between a horizontally polarized fundamental shear mode of oscillation of the outer upper-plate wedge and the oceanward displacement

of this wedge during rupture (Fig. 1A).

In an area ~50 km wide from the trench on the upper plate (Fig. 1B) and ~100 km along it, the static horizontal displacements at the seafloor (Fujiwara et al., 2011) and at the megathrust (Yue & Lay, 2011, 2013) are both about ~50 m. Low rigidity and possible inelastic relaxation of the upper plate (G. Ekstrom personal communication) may account for the lack of attenuation and large size of the displacement. In addition, the first motion includes a dynamic component that can be as much as 30% of

the static displacement (e.g. Yue & Lay, 2011, Fig. 1A). Thus, the initial eastward displacement of the seafloor at the outer margin could have been ~65 m. This seafloor motion is primarily driven by the displacement on the patch of the megathrust below it. This displacement is accomplished in about ~40 seconds and the rupture propagates across the area in our Figure 1B in ~20 seconds (Yue & Lay, 2011). Thus ~1 m/s is a reasonable estimate of the average velocity of the seafloor above the outer wedge during its ~60 seconds first pulse of eastward displacement. This pulse includes the static displacement and a dynamic

overshoot, which is the first cycle of the reverberation that follows. The rupture displacement is expected to drive the upper plate eastward and upward. Assuming that the water column above it moves only upward (e.g. Fujiwara et al., 2011), the relative motion between water and the uppermost sediment should be nearly equal to the motion of the seafloor on the upper plate, producing shear velocities ~1 m/s during the ~1 minute of the first motion. This duration may be extended if peak velocities in the early oscillations can reach similar velocities to the first one (Fig. 1B).

Based on available seismic velocities (Kodaira et al., 2017), we calculate the fundamental periods of horizontally polarized shear waves with nodes at the top of the subducting slab and "crests" at the seafloor (Fig. 1B and S1). These periods tend to increase landward as the upper plate thickens but remain close to ~10 s over a ~30 km wide belt of the outer margin centered on the mid-slope terrace (Fig. 1B). Amplitude and duration enhancements of seismic waves in narrow frequency bands are typical of sedimentary basins, but they have recently been also recognized at active margins by in-situ deep-water

measurements of long-period teleseismic waves (Nakamura et al., 2015; Gomberg, 2018). Lower seismic velocities at outer active margins are usually ascribed to accretion of water-rich sediment and tectonic deformation (e.g. Kodaira et al., 2020). The main purpose of this work is to test whether such long-period motion can develop sufficient water-sediment differential velocity and shear stress to entrain the surficial sediment, and whether high-frequency vertical acceleration can enhance this entrainment (Gomberg, 2018).





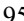

**Figure 1: Profile across the Japan active margin at the 2011 Mw9.0 Tohoku-Oki megathrust rupture. A. Schematic cross section showing hypocenter and rupture initiation (red start) and megathrust domains A, B and C modified from Lay et al. (2012) (no vertical exaggeration). Mean values of rigidity in depth bins from seismic-rupture radiation averaged for a number of megathrusts (Lay and Bilek, 2007). Note order of magnitude increase across the seismogenic range. Red half-arrows show peak co-seismic slip from Lay et al. (2012). Red arrows show the horizontal static displacement from Yue and Lay (2013). B. The outer upper plate above the largest 2011 rupture displacements (based on MCS profile D11). Periods of fundamental-mode horizontally polarized oscillation (red) (Fig. S1). The landward increases in wedge thickness and S-wave velocity (i.e., rigidity; Von Huene et al., 1994; Fujiwara et al., 2011) allow for a ~20km wide plateau in fundamental periods at ~10 sec centered on the Mid-Slope Terrace.**



## 3 Physical experiment setup

**Earthquake Motion**. Our experiments (Text S1) simplify the full spectrum of earthquake motion into two essential components: 1) quasi-steady shear at the sediment-water interface produced by the static displacement and ensuing oscillations of the outer upper plate, which is simulated by steady flow of water over sediment; and 2) high-frequency P-waves (1-10 Hz),
which are simulated by vertical shaking of water and sediment within a rectangular duct (Fig. 2a and 2b).

**Sediment and water Composition**. Mixtures of fine sand, silt and clay-size bentonite simulated sediments. The first series of experiments used two sediment mixtures characterized by a low (10%) and high (40%) fine sand content, with the remainder composed equally of silt and clay, labelled Mix#1 and Mix#2, respectively (Table S1). We investigated the role of water with freshwater content ranging from 50 to 80% by weight (implying 50 to 20% of sediment content in the mixture, respectively).
We also did runs with Mix#3, which had the same dry sediment as Mix#2, but with saline rather than fresh water (Table S1).

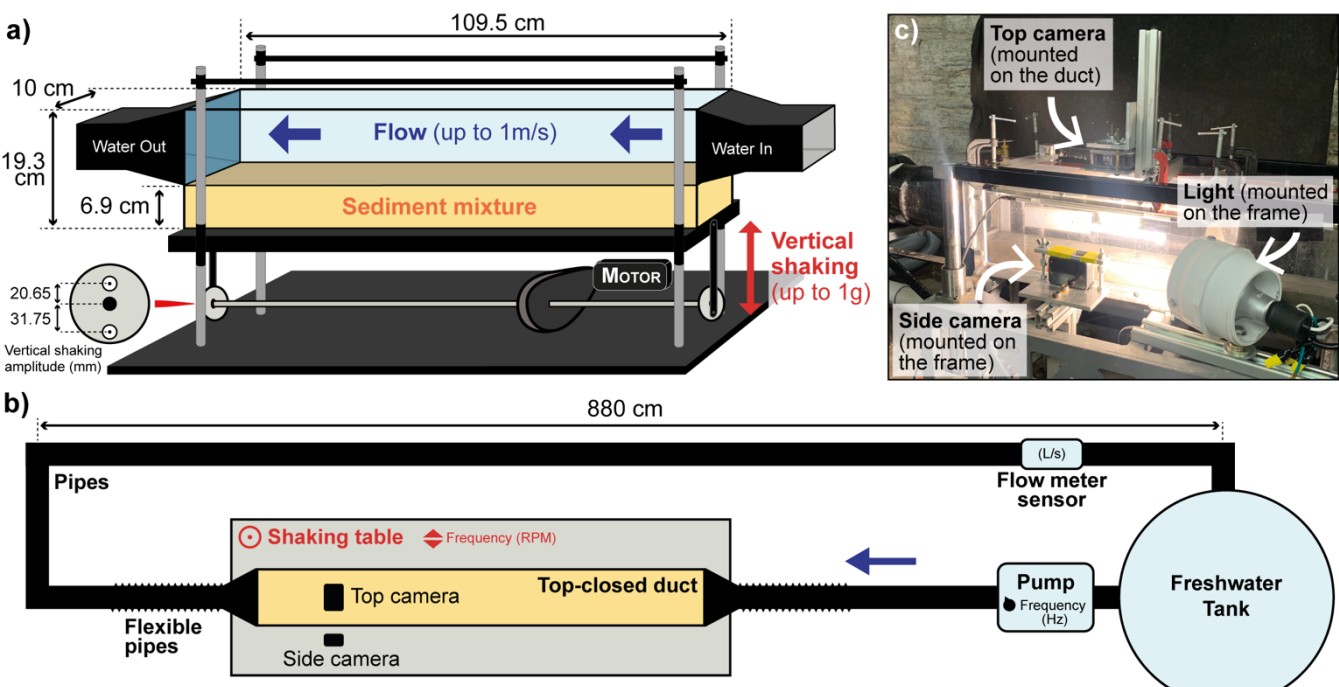

**Figure 2: Physical experiment set up. a) Details of the duct with sediment and water. The shaking table amplitudes are set 20.65 mm or 31.75 mm for each run. b) Top view of the system. c) Photo of the downstream duct area targeted for data recording.**






# 4 Physical experiment results

The first set of experiments compared erosion rates caused by flow in different settings. Mean flow velocity $U$ and vertical peak acceleration $a_v$ were constant in each run, but varied among the runs: 0-1 m/s and 0 or 1g respectively (Fig. 3). We calculated erosion rates $r_v$ in two ways: 1) by differencing bed topography before and after the run; and 2) by measuring from
videos from one side of the duct. Both methods provide consistent results (Table S2). We identified two processes of bed erosion: 1) grain-by-grain erosion; and 2) stripping, characterized by a sudden entrainment of a cm-thick layer of sediment. Once entrained by either mechanism, the sediment is transported out of the duct as bedload and/or suspension.

For sediment Mix#1 and $U$ = 1m/s flow velocity, erosion rates are low, ~0.03 cm/min, with grain-by-grain entrainment, regardless of water content and $a_v$ (Fig. 3). Vertical shaking has no clear effect on entrainment rate even at $a_v$ = 1g. Results for
Mix#2 depend on water content. With 50% and 60% water content and $U$ = 1 m/s, $r_v$ ranges from 0.008 to 0.029 cm/min with predominantly grain-by-grain entrainment. With 70 % freshwater and $U$ = 0.8m/s, we observe two phases (Fig. S2): the first minutes show grain-by-grain entrainment and $r_v$ = 0.08 - 0.18 cm/min, Table S2); then the upper part of the bed is stripped, greatly increasing $r_v$ (1.3 and 1.5 cm/min, Fig. 3 and Table S2). With a flow velocity of 1 m/s, the entire bed is eroded by stripping in 90 s. With 80% water content, $r_v$ for Mix#2 increases to 0.7 - 0.9 cm/min with $U$ = 0.5 m/s (Fig. 3). There is no
major stripping but sediment waves developed at the bed interface. With sea water in the mixture (Mix#3), $r_v$ increases dramatically, to 3 cm/min, even with $U$ = 0.5 m/s (Fig. 3). Finally, for the same mixture and flow velocity, erosion rates are higher when the sediment is subjected to $a_v$ = 1g (runs with 70 % and 80 % water on Fig. 3, Table S2).

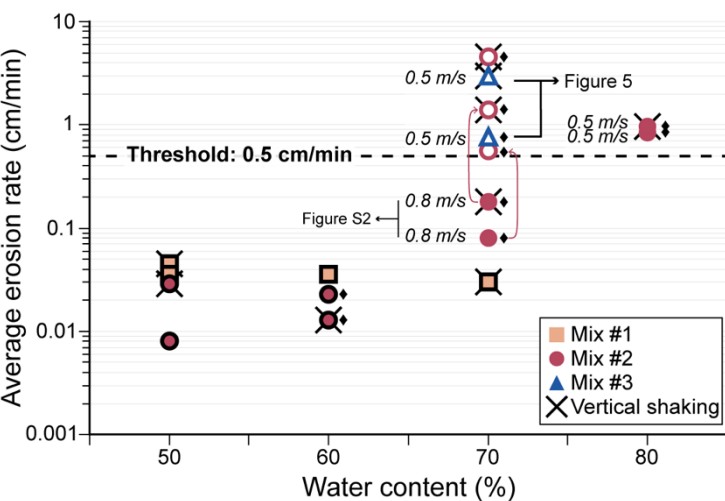

**Figure 3: Entrainment versus sediment water content. Entrainment involves single grains only (full symbol), includes clumps of sediment (black diamonds), as above, but with major rapid stripping (empty symbol). Each dot highlights the mean erosion rate over a run, but arrows link distinct erosion steps in the same run (the symbols at the start of the arrow show the mean erosion rate of the first step and the symbols at the end of the arrow show the mean erosion rate considering both steps). The crosses mark runs with a fixed vertical shaking at 3.5 Hz (with an amplitude of 20.65 mm) and an acceleration of 1 g. No shaking in other runs. The**
**flow velocity was set up at a constant 1 m/s, unless specified. While the black outline indicates that the run lasted 10 min, the other runs were shorter due to the high erosion rates.**




The compared erosion rate versus flow velocity for each of the three mixtures with 70% water content is shown on Figure 4. The runs have no vertical shaking, and a gradual increase of the flow velocity. Values of $r_v$ for Mix#1 and Mix#2 remain low ($<0.1$cm/min) but show distinct trends (Fig. 4A). With Mix#1, erosion starts with $U = 0.35$m/s and increases up to 0.02cm/min at about 0.6m/s, but then remains constant for velocities from 0.6 to 1.0 m/s. With Mix#2, erosion starts with a $U = 0.3$m/s and increases up to 0.08cm/min at 0.8m/s. Figure 4B highlights the significant impact of salinity on entrainment. Erosion rates at $U = 0.5$m/s are 100 times higher for Mix#3 than for Mix#1 and Mix#2.

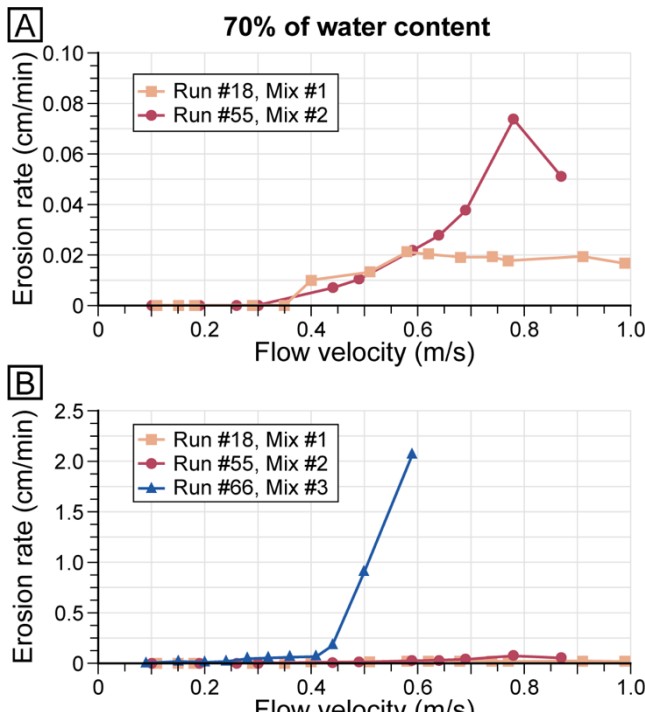

**Figure 4: Evolution of the erosion rate for 3 representative runs (one for each mixture with a water content of 70 %) with no vertical shaking and increasing the flow gradually. The plots A and B show the same curves for the Mix#1 and #2, but with an expanded vertical scale for the Mix#3 run.**

A closer look at the evolution of erosion rates over time for the Mix#3 runs shows the same trend with or without vertical shaking. During the first seconds, the sediment is eroded by stripping, leading to $a_v$ values up to 13 cm/min and 10.5 cm/min for runs 67 and 69, respectively (Fig. 5). In a second phase, the erosion occurs grain-by-grain and via sediment clumps, with strong influence by the vertical shaking. With $a_v = 0$, $r_v$ ranges between 0.2 and 0.9 cm/min (Run 67, Fig. 5) and between 1.8 and 2.9 cm/min for $a_v = 1$g (Run 69, Fig. 5).





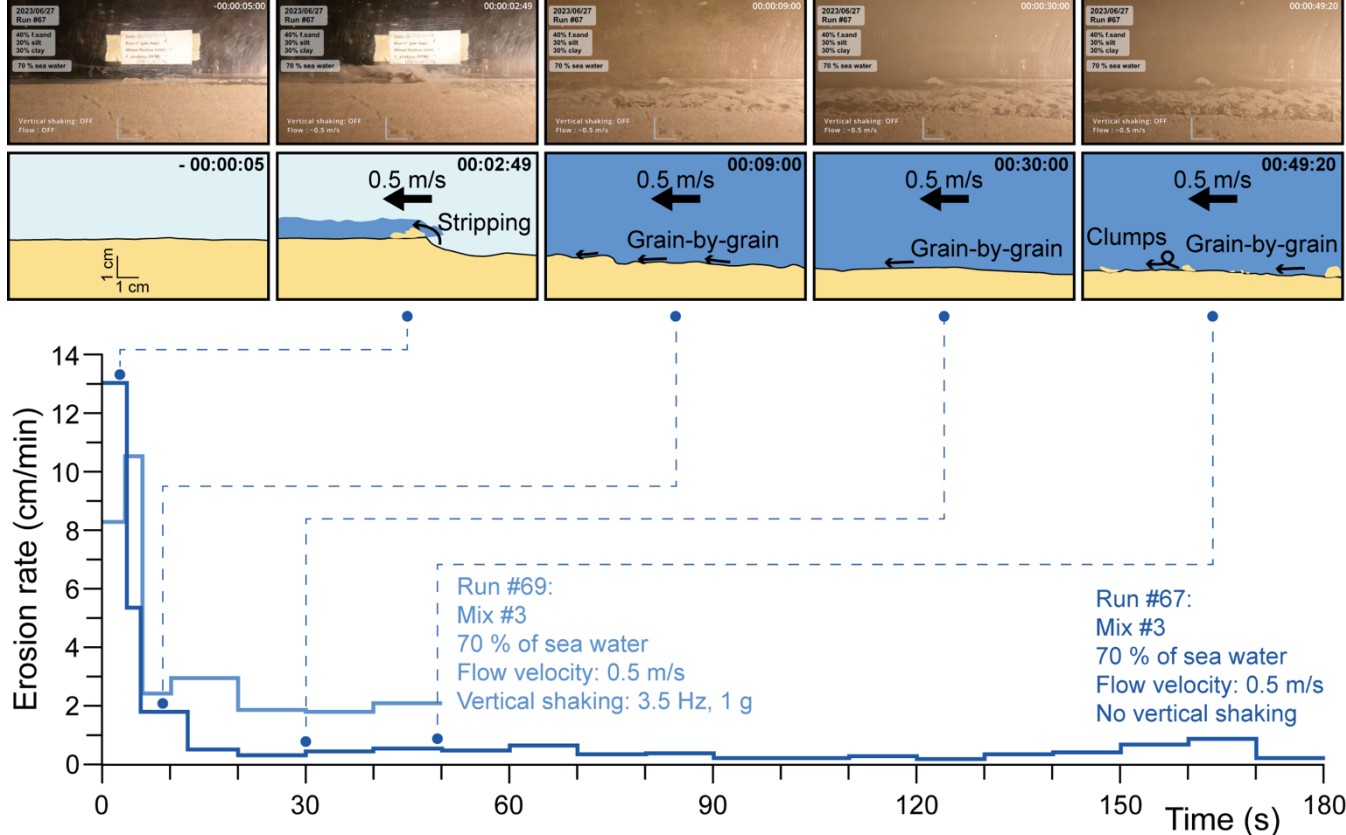

**Figure 5: Evolution of the erosion rate of Mix#3 sediment. The erosion rates are calculated over 10 s-intervals but more often for**
**the first ~10 seconds of each run. The photos and their interpretative cartoons highlight different steps of run #67.**

## 5 Discussion and Conclusion

The initial experimental results we report here show that the seafloor motion of the outer upper plate in response to the 2011
M9.0 rupture could cause widespread entrainment of surficial sediment with grain sizes and water contents consistent with the
abyssal floor. Generally, our laboratory studies distinguish modes of remobilization that result from the interplay of high-
frequency and low-frequency motions. They show that relative shear between the seabed and water column compatible with a
large rupture can effectively entrain surficial sediment and lead to the types of event deposits created by the Tohoku-Oki
earthquake. The short-lived radioisotopes measured from the submarine deposits demonstrated that the Tohoku-Oki earthquake
triggered remobilization of the upper few centimeters of sediment over a wide area of the seafloor (McHugh et al., 2016).
Based on this thickness and considering a duration of several minutes for the strong motion, including rupture and reverberation
(Nakamura et al., 2015; Gomberg, 2018), we propose a threshold $a_v$ value of ~0.5 cm/min, above which the erosion rate of
surficial sediment is consistent with the field data (Fig. 3).



Our investigation of the surficial sediment entrainment under controlled conditions highlights that the intensity of the seismic waves is not the only factor. The composition and the physical properties (e.g. water content, shear strength) of the sediment also strongly influence its susceptibility to entrainment. The results obtained with Mix#2, enriched in sand-size particles, show

the impact of the water content on entrainment. Entrainment by stripping occurs in Mix#2 with ≥70% water, whereas erosion is insignificant below it (Fig. 3). In the field, the water content of the sediment decreases with increasing sediment burial due to compaction, typically reaching about 50% at a depth of 10 - 20 cm. Thus, entrainment of surficial sediment by long-period motion is likely to be limited by the depth to more compacted sediment. Clay concentration and the ionic strength of the interstitial water (freshwater or sea water) are also significant factors (Fig. 4). Other parameters that could affect sediment

entrainment include components like diatoms, ashes, organic matter biofilms, and also the nature and intensity of bioturbation. The long-period motions (i.e. the quasi-steady flow in our experiments) can effectively remobilize relatively weak sea-floor sediment on its own. But, for relatively weak, water-rich sediments, entrainment is enhanced by strong high-frequency vertical motions (PGA = 1g) (Figs. 3 and 5). Our experiments do not include high-frequency shear wave motions, which might also contribute to remobilizing sediment.

### Acknowledgements


This work was funded by National Science Foundation OCE-2044915. We would like to acknowledge the support of Erik Steen, Chris Milliren and Erik Noren in the development of the experiment, and John Leeman from Leeman Geophysical for the construction of the shaking table.

### Data availability

The data used for this manuscript are the recording videos of each run. Reduced-resolution copies of the videos are available at the following archival identifier: https://hdl.handle.net/11299/263944.

### Competing interests

The authors declare that they have no conflict of interest.

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
