# Peer review of "Supporting information for"

_EGUsphere, 2024_

## Author Comment (AC1)

**Joan gomberg's comment**

This paper presents results of a novel experiment that is well thought out and interesting.  It should set the precedent for subsequent studies, which is what good science should do! The rationale for the work and the design of the experiment is particularly well explained, and over all the paper is well written.  I believe that with some minor clarifications the paper should be publishable.

I suggest that in addition to the emphasis on the 'tsunamigenic' aspects of subduction thrust earthquakes (e.g., as in line 19 stating "Despite their high tsunamigenic risk") the authors also note the hazards associated with the shaking and consequent damage, which lead to significant risks.  This is particularly true given the long duration and low-frequencies of the shaking that are unique to these large megathrust earthquakes and impact large major structures (e.g., tall buildings, long bridges, submarine cables) and ground failure events (e.g., liquefaction, landsliding).
We agree. We further emphasize the risk associated with a major megathrust earthquake in L19-20.

A more complete test of the hypotheses tested in this study and their significance would include demonstration that only waves with characteristics unique to large earthquakes mobilize surficial sediments; i.e., in addition to showing that long period oscillations can mobilize surficial sediments it would be useful to know that shorter period oscillations do NOT do so! While the experiments were not designed to test a range of frequencies, even just noting more clearly what published studies show would be useful.  That is, are there observations of traditional turbidites from M<9 earthquakes that did NOT mobilize surficial sediments?
Most papers until McHugh et al (2016) focused on the sediment transport induced by high-frequency shaking. The surficial nature (upper few cm's) of sediment transport was proven by short-lived radioisotope dating over large spatial distances identifying a new sedimentation processes. This was facilitated by all that is known from the Mw9.0 Tohoku earthquake.

The paragraph between lines 71-84 would benefit from some revision, to make it more concise and clearer.  I have tried to make some suggestions in the annotated text.
The conceptual model section has been re-organized to make it clearer. We thank the reviewer for her suggestions which were very helpful in doing that.

Please describe what the physical rationale is for assuming that high-frequency vertical acceleration enhances entrainment, as suggested in lines 93-4?
We have modified this sentence (now L50) by removing the assumption and replacing it with a question asking what will be the consequence of the interaction between long-period motion and high-frequency vertical acceleration.

Perhaps it would be clear to a sedimentologist, but for a non-sedimentologist like me the description of the sediment water compositions in lines 111-115 needs clarification

(and should be clear without having to look at the Supplement for Table S1). There seem to be two fine sand contents (percentages are fractions relative to what?), and 50% to 20% sediment content. How do these percentages relate to the fine sand percentages? The phrase "with the remainder composed equally of silt and clay" is unclear; what is this the remainder/leftover of, does "equally" mean the same amounts each of silt and clay or something else, and how is amount measured? How do all these things relate to just two mixtures, Mix 1 and Mix 2?

We have re-written the whole section to make it clearer. For each percentage, we've explained what it relates to, to avoid confusion.

I would also suggest calling the three mixtures by names that are more descriptive than just numbers, so the reader doesn't have to remember the characteristics implied by the numbers. For example, instead of Mix 3 it could be called the 'saline XXX mix', and Mix 2 the "fresh XXX mix" with XXX noting the distinguishing feature of Mix 2.

We have renamed the mixture: fresh-sand-poor mixture; fresh-sand-rich mixture; and sea-sand-rich mixture.

**PDF comments**

L33: Insert "it"

We corrected the sentence.

L33-34: I'm not sure this is really the appropriate reference. If this paper mentioned sediment entrainment and the mechanism for it, the later was purely speculative!

We especially appreciate this comment given that the reviewer is referring to her own paper. And we understand that this mechanism is not the topic of Gomberg (2018). However, this is the first time that it is mentioned in the literature, and we would prefer to leave the reference. We re-wrote the sentence to be more accurate by saying "as suggest by Gomberg (2018).

L42: what mechanism? The previous sentence describes signatures, not mechanisms?

We replaced the word "signatures" by "processes" on L31, and "mechanism" by "process" on L36.

L44-45: These seem to be the key observations that require large-scale shaking, as simply mobilizing shallow sediments seems plausible done locally by smaller earthquakes too (e.g., if shaking caused more deep-seated slope failures, that dump sediment into the water column locally, wouldn't this sediment include the deeper sediments plus the shallow ones atop them?). I.e., it's the thickness and widespread nature of the sediment source that ties them to a big earthquake, not that their sourced from surficial sediments?

To explain the isotopic composition of the deposits described in McHugh et al. (2016) and associated to the Tohoku earthquake, the authors argue that the deposits are the result of entrainment of only surficial sediment. In order to explain the thickness of these deposits, this remobilization would have to occur over a very large area, beyond the rupture zone of the earthquake. McHugh et al. (2020) inferred this co-seismic

surficial sediment remobilization to be due to the co-seismic long-period motion, which could only be triggered by tsunamigenic earthquake.

L59: remove "entire brittle", "erosional" and "from its nucleation, this rupture propagated both"
We removed "entire brittle" and "from its nucleation, this rupture propagated both" from the sentence, however, we left "erosional" but we added as reference Von Huene et al. (1994).

L68: change to 'also'
We corrected the sentence.

L71: ??
The 50 by 100 km zone we refer to in this section is the area where the slip was the largest during the Tohoku earthquake. We have rewritten the sentence to make it clearer.

L72-73: I'm a bit confused by this … attenuation is usually attributed to inelastic relaxation, diminishing elastic seismic waves, so it seems odd that this would be a reason for low attenuation?
The new version doesn't include this. The wrong word was used in the previous version.

L74: is this the dynamic overshoot mentioned in lines 79-80?
We modified this sentence to clarify.

L77: which displacement (previous sentence mentions seafloor and megathrust displacements)? Maybe refer to the motion along the megathrust as 'slip' and seafloor as 'displacement'?
This is a great way to clarify the information. We followed this advice.

L78-79: This is not necessary here, as the relevant velocities are noted in lines 82-3.
We agree that it was a repetition of information. We followed the advice of the reviewer.

L80-81: Replace with "The largely eastward megathrust slip and seafloor displacement also causes deformation and uplift of the sediments,"?
We removed this sentence.

L86: You could simplify this sentence by omitting this phrase and changing "fundamental periods" to "fundamental resonance periods".
We made the change and moved the sentence into the caption of Figure 1B.

L124: what does the subscript 'v' refer to?
The subscript 'v' substant to "vertical". We left it as it is for the vertical acceleration $a_v$, but we removed it from the erosion rate, which is now only $r$.

L144: insert 'applied'?
We corrected the sentence.

L159 : should be 'rv'?
Yes, it should. We corrected this.

L175 : should be a 'flow rate' instead?
This should be the erosion rate. We corrected it.

L178: add "determining whether sediment may be mobilized"?
We did it.

**Michael Clare' comment**

This is an interesting, well-written and well-illustrated study that performed experiments to understand the response of sediments to disturbance by large magnitude mega-thrust earthquakes. It provides an improved mechanistic understanding of a previously observed thin remobilisation of surficial seafloor sediments following large quakes in several sites. It is therefore a novel and widely-relevant study.

Scaling – some discussion on scaling issues/limitations of the experiment should be added. Including the issues of only considering one type of clay (bentonite). What are the key assumptions or uncertainies in scaling this up to the real world?
We appreciate the chance to address this fundamental question for an experimental study. Our physical experiments are focused on entrainment at the sediment water interface. They are simplified in some respects, but the key processes – boundary layer flow, sediment entrainment, and vertical shaking – are present at their natural scale in the experiments. They are designed to investigate whether a shear stress induced at the water/sediment interface can entrain marine sediment, and measure the consequences of its interaction with high-frequency shaking.
For the experiments presented in this manuscript, we have used only one type of clay, as this is a first-order stage in our investigation of physical experiment results. The aim of future works will include exploring the impact of different mixture parameters that may affect the sediment entrainment (for instance the clay mineralogy). This is specified at the end of the discussion/conclusion section.
We used one type of clay that is most common on the Japan Trench margin given that the process was identified in this margin, but other clays will be used for experiments in the future.
To be clearer on our intention in this manuscript, we added in the introduction (L55-56) that it is the initial report of this physical experiment investigation and we specified in the supplementary materiel text (which described the physical experiment), that the experimental system is not a scale model and that the sediment mixture is simplified but will be made more "realistic" for future experiments.

The Introduction should also address other mechanisms that can mobilise sediments during/after earthquakes as this is not the only mechanism that has potential to occur.

We have added a sentence, L44, to insist on the fact that we are investigating in this work a newly co-seismic mechanism for sediment entrainment during earthquake that would take its place along with other known co-seismic processes capable of demobilizing sediment.

The fidelity of palaeoseimology is painted as being complete in the introduction, so I think it is necessary to recognise that this is not always so straightforward and will not apply everywhere. Not much needs to be added, but just a recognition that there are places where there may not always be a direct link between large earthquake and resultant deposit.

We agree that submarine paleoseismology is a challenging field. While we do not agree that we implied it was "complete" in our original version, we modified the sentence (L25) to highlight this.

Add to the conclusions/discussion the wider application of this study. Where is it most likely to be applicable? And where will it not be? This would be useful addition information for the reader.

The goal of our manuscript is to introduce a new fundamental mechanism of co-seismic entrainment. We provide a context for this new mechanism, and as far as we know, it could happen for any earthquake we described. However, providing a detail seismological context is beyond the scope of our paper.

Line 26 – I would caveat this to add some caution around the fact that this is not always possible and records can have varying degrees of fidelity. As-written this reads as if turbidite paleoseismology is always valid so would suggest some balance.

Howarth, J.D., Orpin, A.R., Kaneko, Y., Strachan, L.J., Nodder, S.D., Mountjoy, J.J., Barnes, P.M., Bostock, H.C., Holden, C., Jones, K. and Cağatay, M.N., 2021. Calibrating the marine turbidite palaeoseismometer using the 2016 Kaikōura earthquake. *Nature Geoscience*, *14*(3), pp.161-167.

Bernhardt, A., Melnick, D., Hebbeln, D., Lückge, A. and Strecker, M.R., 2015. Turbidite paleoseismology along the active continental margin of Chile–Feasible or not?. *Quaternary Science Reviews*, *120*, pp.71-92.

Nieminski, N.M., Sylvester, Z., Covault, J.A., Gomberg, J., Staisch, L. and McBrearty, I.W., 2024. Turbidite correlation for paleoseismology. *Geological Society of America Bulletin*.

Atwater, B.F., Carson, B., Griggs, G.B., Johnson, H.P. and Salmi, M.S., 2014. Rethinking turbidite paleoseismology along the Cascadia subduction zone. *Geology*, *42*(9), pp.827-830.

Maier, K.L., Strachan, L.J., Tickle, S., Orpin, A.R., Nodder, S.D. and Howarth, J., 2024. Testing turbidite conceptual models with the Kaikōura Earthquake co-seismic event bed, Aotearoa New Zealand. *Journal of Sedimentary Research*.

We have modified the sentence (L25) to emphasize that submarine paleoseismology remains a challenging method. We have also added Talling (2021) reference which states this.

Line 31 - Understanding event deposits starts with sediment entrainment.

I would suggest that that it is not just entrainment but also remobilisation. I think the study should introduce early on the potential mechanisms that can involve mobilisation

of sediments during or after an earthquake to set the scene effectively, as it currently reads as if there is only one possible mechanism.

For example, direct ground motion mobilising thin surficial fluid-rich sediments, cyclic loading that destabilises continental slope sediments that generate a mass movement (landslide, debris flow etc), generation of sediment density flow due to disturbance of sediment into the water column, as well as secondary events such as earthquake-triggered tsunami that disturbs seafloor sediments. This would be useful context up front. The mechanism proposed by the authors is valid but is not the only mechanism that gives rise to deposits that may potentially be linked to earthquakes.

We have listed all the co-seismic submarine sedimentary mechanisms which occurred along the Japan subduction zone associated with the Tohoku-oki earthquake in order to highlight this variety of processes that can be triggered by an earthquake. L43-45, we added a sentence to specify that in this manuscript, we are focusing on a new process which has only been briefly described so far in a few publication (Gomberg, 2018; McHugh et al., 2020).

General question on the experiment setup: To what extent does bentonite represent real-world conditions? There is quite a range of cohesivity across different types of clays so some explanation or justification of bentonite would be useful.

Using only bentonite for the clay component of the mixture is not representative of real-world conditions, but it is representative of a relatively reactive clay mineralogy that certainly occurs in nature. We have presented in this study the first step using a new physical experiment device. Even if we cannot exactly reproduce submarine sediment by making mixtures in the lab, our mixtures have size distributions and water contents that are generally representative of natural marine sediments, which of course are highly variable. Future experiments will aim to use more realistic mixtures.

Line 125 – these two mechanisms are interesting. I think setting up the different mechanisms that have been proposed by past studies for this remobilisation would be useful to introduce in the Introduction. The reader really needs to know what the current state of knowledge/understanding is regarding these mechanisms to determine whether the results are confirmatory or really novel.

Grain-by-grain erosion and stripping are not a new way to entrain the sediment, but the circumstance of this entrainment is unique. We added in the manuscript (L126) a sentence to make clear that these two mechanisms are already known with this type of sediments.

Figure 4 – I would find it useful if the details of the mix, as well as the name of the Mix could be added to the legend so it is really clear to the reader and the figures can stand-alone.

We agree that the labels of the mixtures were not helping to read the manuscript. We changed their names in the text, as well as on the figures and tables, in order to be straightforward: fresh-sand-poor mixture, fresh-sand-rich mixture and sea-sand-rich mixture.

Line 167 – this confirms previous hypotheses from a number of studies in Japan and elsewhere (e.g. Moernaut et al. 2017) - but this new study provides a mechanism to explain this. I would suggest broadening the study reach and also referencing some of the studies from elsewhere too.

Moernaut, J., Van Daele, M., Strasser, M., Clare, M.A., Heirman, K., Viel, M., Cardenas, J., Kilian, R., de Guevara, B.L., Pino, M. and Urrutia, R., 2017. Lacustrine turbidites produced by surficial slope sediment remobilization: a mechanism for continuous and sensitive turbidite paleoseismic records. *Marine Geology*, *384*, pp.159-176.

Moernaut et al. (2017) described surficial sediment entrainment process in Chilean lakes, as McHugh et al. (2016) documented it along the Japan trench. However, these two studies do not propose the same co-seismic mechanism as trigger of this entrainment. Here, we are focusing on the hypotheses raised by McHugh et al. (2020), which involve a shear stress due to the significant co-seismic seafloor motion. We show in our study that such motion can only be triggered by tsunamigenic megathrust ruptures.

**Valerie Sahakian's comment**

This is a really interesting and cool study that is an important contribution to understanding the physical processes behind our observations of paleo event deposits. I am excited to see it published, and have a handful of comments below that I think could strengthen the work, and links to interpretations of deposits moving forward.

Main comments:
- Because the motivation/interpretation that entrainment is related to long period motions, and because this relies on assumptions of the likely frequencies that will be amplified in different portions of the wedge, I think it is important to expand your description of how you attain the frequencies in Figures 1 and S1; perhaps move the text from the caption of Figure S1 to its own supplementary section, and expand on the equations used to go from 2.1km depth -> 6s fundamental mode; it is not entirely clear to me from the text how this was obtained. A table might support this, with average Vp or Vs velocities and depths

We would like to emphasize that the motivation of our work is testing whether the long-period components of the co-seismic motion can account for the entrainment of surficial sediment as implied in the event deposits described by McHugh et al. (21016). We made this clearer in the introduction (L48 to 50). Regarding the calculation of the fundamental periods of horizontally polarized shear waves, we took the advice of the reviewer. We made a supplementary materiel 2, which consist of a text (previous version of the figure's caption), the same figure but with a more adapted caption, and table which support our calculations.

- Given that these fundamental modes are likely amplification of horizontal motions, it would be good to discuss in the manuscript how this is accommodated with your experimental setup, and how that contributes to your interpretation that it is long period motion that generates entrainment. i.e., is it from the water flow? If so, how do you relate flow velocity to amplification at

those specific periods? The vertical motions in the experiment are higher frequency than the oscillatory modes predicted, so I think this component is important to link the horizontal component of your setup to static or dynamic deformation/ground motions, and how your experiments translate to existing observations (or how the could/should be modified moving forward to do so).

We have rewritten the physical experiment set up description to clarify which part of the seismic motion the flow and the vertical shaking of the experiment aim to mimic (L 106 to 109).

The flow over the sediment in the experiment produces the same shear stress at the water-sediment interface as that expected in the field from co-seismic long-period motion of the bed. The shear stress is the same if the bed is still and the water is moving or vice-versa.

In our conceptual model, we offer a quantitative estimate of the motion at the seafloor of the outer wedge during the mainshock rupture based on published results of the motion at the megathrust and of the static displacement at the seafloor. We obtained the initial average shear velocity between sediment and water of ~1 m/s, which is a constraint we adopted for our experiments.

The discussion on the fundamental mode points to a likely resonance between these modes and the seismic motion imposed on the outer wedge by the large earthquake rupture. Our calculation derives the frequencies of the fundamental modes along the profile in Figure 1.B. Despite the simplistic assumptions, the result shows that the frequencies vary little over a substantial width of the wedge and therefore may oscillate in unison and be more likely to develop large amplitude. But we do not calculate the amplitudes, which requires dynamic modeling beyond the scope of this paper.

While the static displacement at the seafloor might occurs in about 60 second, this resonance effect may add significantly to the shear.

We have rewritten the conceptual model section to make it clearer.

- Lastly, I think it would be good/important to show a Figure 4, but plotting all runs, and add in the supplement, to represent all data and how they contribute to the interpretation (or why certain runs were selected for the final interpretations but not all)

The numeration of each run is derived from the list of experiments conducted over the course of a month-long session working on the physical experiment. Many of these runs were exploratory in nature. In this manuscript, we present and analyze all the runs that are directly comparable, providing a focused discussion of their outcomes and relevance.

Minor comments:
- Suggest changing Mw to a bold M (**M**); Mw is technically work magnitude and is used because of a misinterpretation of some Hanks and Kanamori publications (Tom Hanks was clear that moment magnitude is supposed to be a bold M).

We understand that nomenclature is the subject of debate, but we have chosen to follow the one, which is Mw.

- It would be useful to have references to support this statement on line 31: "The premise in sediment dynamics has been that the bed is fixed, and the fluid moves relative to it, creating shear stress that entrains sediment. "

We want to acknowledge that it is a good point. We would like to add a reference that highlight this statement, however, everything in fluid dynamic starts with this, that the bed is fixed and the fluid above move. We can't pick only one reference, but if the reviewer insists about it, we will add some examples.

- It would be useful in the Introduction to specify what "low frequency/long-period" is (i.e., <1Hz, <0.1Hz, etc.). e.g., lines 48-55

The long-period waves are defined in Figure 1B by period values of ~10s.

- Would be good to clarify this statement on lines 71-72, that it is specific to Tohoku: "In an area ~50 km wide from the trench on the upper plate (Fig. 1B) and ~100 km along it, the static horizontal displacements at the seafloor (Fujiwara et al., 2011) and at the megathrust (Yue & Lay, 2011, 2013) are both about ~50 m"

We clarified in the text that this displacement is specific to Tohoku rupture (L 69 to 70).

- Line 73: An appropriate reference in place of Ekstrom personal communication would be Ma 2012, GEOPHYSICAL RESEARCH LETTERS, VOL. 39, L11310, doi:10.1029/2012GL051854, 2012 .

We thank the reviewer for this useful reference. However, we have removed this statement in the new version of the conceptual model.

- Clarify line 85: "Based on available seismic velocities" – clarify Vp, Vs, etc.

We specified that we use Vp model from Kodaira et al., 2017 (L84).